# Covariance-corrected Whitening Alleviates Network Degeneration on Imbalanced Classification

## Abstract

Class imbalance is a critical issue in image classification that significantly affects the performance of deep recognition models. In this work, we first identify a network degeneration dilemma that hinders the model learning by introducing a high linear dependence among the features inputted into the classifier. To overcome this challenge, we propose a novel framework called Whitening-Net to mitigate the degenerate solutions, in which ZCA whitening is integrated before the linear classifier to normalize and decorrelate the batch samples. However, in scenarios with extreme class imbalance, the batch covariance statistic exhibits significant fluctuations, impeding the convergence of the whitening operation. Therefore, we propose two covariance-corrected modules, the Group-based Relatively Balanced Batch Sampler (GRBS) and the Batch Embedded Training (BET), to get more accurate and stable batch covariance, thereby reinforcing the capability of whitening. Our modules can be trained end-to-end without incurring substantial computational costs. Comprehensive empirical evaluations conducted on benchmark datasets, including CIFAR-LT-10/100, ImageNet-LT, and iNaturalist-LT, validate the effectiveness of our proposed approaches.

## 1 Introduction

In the real-world recognition applications, long-tailed label distribution (*i.e.*, imbalanced datasets) is a common and natural problem, where a few categories (*i.e.*, head classes) have more samples than others (*i.e.*, tail classes). This challenging task has received increasing attention in recent years Cui et al. (2019); Cao et al. (2019); Kang et al. (2020); Zhou et al. (2020); Menon et al. (2020). In previous literature, the methods can be roughly categorized into three groups: re-sampling-based Chawla et al. (2002); Huang et al. (2016); Zang et al. (2021), cost-sensitive re-weighting-based Cui et al. (2019); Cao et al. (2019); Shu et al. (2019); Menon et al. (2020) and other methods Kang et al. (2020); Zhou et al. (2020); Zhang et al. (2019); Zhong et al. (2021). To improve the classification accuracy of tail classes, the re-sampling approaches change the sampling frequency to balance the label distribution, and the re-weighting approaches allocate large weights for tail classes via the loss function, thus an unbiased classifier can be learned.

In this paper, we explore the question of *what causes the poor performance of end-to-end Experiential Risk Minimization (ERM) model training for the imbalanced classification*. To answer the above question, we investigate the feature representations in the hidden layers learned in end-to-end training. As shown in Figure 2, we find that the features fed into the classifier are always highly correlated when the ERM model is trained on imbalanced datasets. This representation collapse makes the training intractable and finally leads to degenerated models. Prior works LeCun et al. (2012) demonstrated that the good features should be decorrelated and have same covariances to avoid producing degeneracies. To this end, we propose a simple yet effective end-to-end training framework named Whitening-Net integrating the whitening transformation into the model to decorrelate the features before being fed into the classifier, which can scatter the batch samples and thus avoid the representations collapse to a compact latent space. Notice that on imbalanced classification, the mini-batch covariance statistic could be unstable, which can result in the whitening operation not converging. Therefore, we propose two covariance-corrected modules, Group-based Relatively Balanced Batch Sampler (GRBS) and Batch Embedded Training (BET) to get more accurate and stable batch statistics to reinforce the capability of whitening. The extensive empirical results on the benchmarks

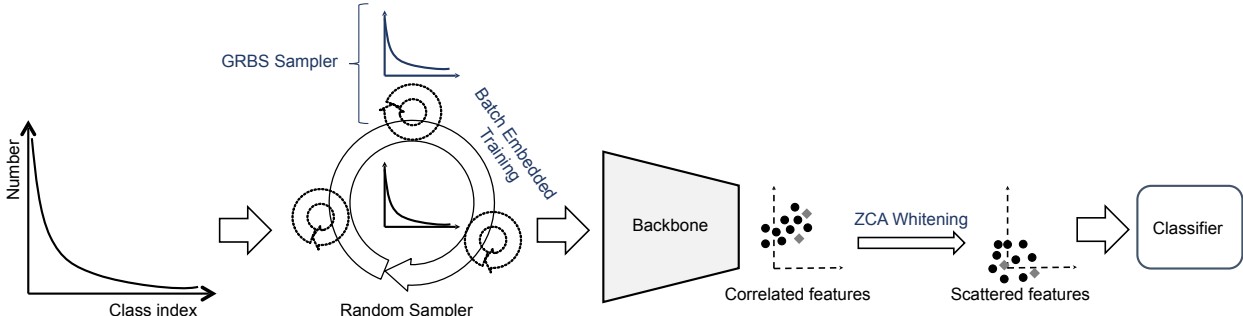

Figure 1: The proposed end-to-end training framework for imbalanced classification. The proposed system includes ZCA whitening on the features before being fed into the classifier, and two covariance-corrected modules, Group-based Relatively Balanced Batch Sampler (GRBS) and Batch Embedded Training (BET).

CIFAR-LT-10/100, ImageNet-LT and iNaturalist-LT demonstrate that our framework can effectively escape from the degenerated models.

The main contributions of this paper are summarized as follows:

- We identify that the highly correlated features fed into the classifier makes the failure of end-to-end ERM model training on imbalanced classification. To avoid feature representation collapse, a simple yet effective end-to-end training framework Whitening-Net is proposed to decorrelate the features.

- Two covariance-corrected modules, Group-based Relatively Balanced Batch Sampler (GRBS) and Batch Embedded Training (BET), are designed to obtain more accurate and stable batch statistic estimation for whitening to avoid its non-convergence and reinforce its capability in imbalanced scenarios.

- Extensive quantitative and qualitative experimental results on four imbalanced benchmarks demonstrate effectiveness of our proposed method. In addition, our approach adds only a very small inference cost.

## 2 Related Works

In this section, we firstly review some representative works on imbalanced classification, including re-sampling, re-weighting and decoupled training methods. Next, some applications of whitening in neural networks are introduced.

**Re-sampling.** Re-sampling methods as one of the classical approaches include over-sampling Chawla et al. (2002); Han et al. (2005); Singh & Dhall (2018); Yan et al. (2019) for the tail classes, under-sampling Kubat et al. (1997); Drummond et al. (2003); Tahir et al. (2012); He & Garcia (2009); Huang et al. (2016); Zang et al. (2021) for the head classes, and heuristic re-sampling Pouyanfar et al. (2018). Although promising results are reported repeatedly in the literature, they still have their own limitations. To be precise, the over-sampling methods augment the tail classes by duplicating samples and they could result in over-fitting Kubat et al. (1997); Drummond et al. (2003); Tahir et al. (2012) onto the tail classes. The under-sampling methods randomly discard some samples of head classes, leading to poorer generalization ability Chawla et al. (2002); Han et al. (2005); Singh & Dhall (2018). Therefore, in recent years, re-sampling methods have fallen out of favor, and the mainstream focused on how to combine different re-sampling approaches on two training states, i.e., learning the backbone and fine tuning the classifier, to learn better classifiers Kang et al. (2020); Zhou et al. (2020); Zhang et al. (2019).

**Re-weighting.** Re-weighting methods Khan et al. (2017); Lin et al. (2017); Zhang et al. (2017); Ren et al. (2018); Pang et al. (2019); Cui et al. (2019); Khan et al. (2019); Cao et al. (2019); Shu et al. (2019); Tan

et al. (2020b;a); Jamal et al. (2020); Ren et al. (2020); Menon et al. (2020) usually allocate large weights for training samples from the tail classes in the loss functions to learn an unbiased classifier. Cui et al. (2019) proposed to adopt the effective number of samples instead of proportional frequency. Thereafter, Cao et al. (2019) explored the relationship between the margins of tail classes and the generalization error and designed a label-distribution-aware loss to encourage a larger margin for tail classes. Balanced Meta-Softmax (BALMS) Ren et al. (2020) proposed an extended margin-aware learning method. Menon et al. (2020) proposed a "logits adjustment" approach by reducing the logits value based on the label frequencies. However, these methods have a large performance gap compared with the following decoupled training methods.

**Decoupled training.** Kang et al. (2020) proposed a decoupled training strategy to disentangle representation learning from classifier learning and achieved surprising results. Zhou et al. (2020) proposed a unified Bilateral-Branch Network (BBN) and a cumulative learning strategy to gradually switch the training from feature representation learning to the classifier learning. Similar work is that Zhang et al. (2019) proposed to integrate two sampling approaches, i.e., random sampling and class balanced sampling, with a feature extraction module and three classifier modules respectively to balance the feature learning and classifier learning. These decoupled training methods achieve better performance than the re-weighting ones which adopt the end-to-end training scheme. The potential limitation of decoupled training is that it cannot search the model globally in the whole hypothesis set and would generally lead to sub-optimal solution. In this paper, our proposed framework can be trained end-to-end to find better feature representations.

**Whitening.** Whitening Koivunen & Kostinski (1999) is a linear transformation that transforms data into a distribution with the mean being zero and the covariance matrix being the identity matrix. After whitening, the features become uncorrelated and each of them has the same variance. Whitening is always used as a preprocessing method Kessy et al. (2018) in real tasks. In recent years, whitening has been introduced into deep neural network applications, including normalization Huang et al. (2018); Pan et al. (2019); Huang et al. (2020b), generative adversarial networks Siarohin et al. (2018), and self-supervised learning Ermolov et al. (2020). In this work, we are the first to show that whitening can be used in long-tailed classification to avoid feature representation collapse and enable the end-to-end training scheme to achieve better performance than the decoupled approaches.

## 3 Method

In this section, we first analyze the network degeneration dilemma on imbalanced classification by visualizing feature distributions. Next, we propose a simple yet effective framework based on channel whitening to normalize and decorrelate the representations of the last hidden layers. Finally, two covariance-corrected modules are proposed to avoid non-convergence and reinforce the capability of whitening in imbalanced scenarios via obtaining more stable and accurate batch statistic estimations.

Let $\mathbf{X} = [\mathbf{x_1}, \mathbf{x_2}, ..., \mathbf{x_C}]^{\mathrm{T}} \in \mathbb{R}^{C \times B}$ be a mini batch of features before being fed into the classifier, where $C$ and $B$ are the number of channel and batch size respectively.

### 3.1 Network Degeneration on Imbalanced Classification

This section aims to visualize correlation coefficients among the channel-wised feature representations in the hidden layers to identify the key factor causing the failure of end-to-end training scheme on imbalanced classification.

In order to analyze the linear correlation among the channels of $\mathbf{X}$, we compute their pearson product-moment correlation coefficient (PPMCC) by:

$$\rho(x_c, x_{c'}) = \frac{\sum_{i=1}^{B}(x_{ic} - \bar{x}_c)(x_{ic'} - \bar{x}_{c'})}{\sqrt{\sum_{i=1}^{B}(x_{ic} - \bar{x}_c)^2}\sqrt{\sum_{i=1}^{B}(x_{ic'} - \bar{x}_{c'})^2}} \tag{1}$$

where $c, c' = 1, 2, ..., C$. Thus, the value of $\rho$ can vary between $-1$ and $1$. The larger absolute value $|\rho|$ means that the channels $x_c$ and $x_{c'}$ are more linearly correlated.

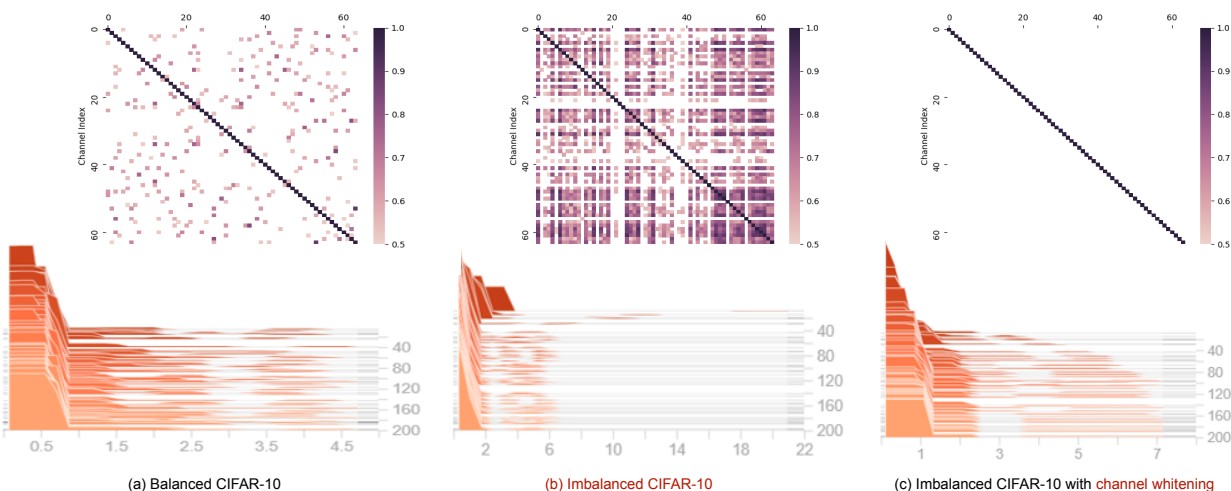

(a) Balanced CIFAR-10   (b) Imbalanced CIFAR-10   (c) Imbalanced CIFAR-10 with channel whitening

Figure 2: The visualizations of feature distribution before being fed into classifier. The top row figures show the correlation coefficients between channel-wised features. The bottom row figures illustrate the singular value histograms of features. The X-axis represents the singular value, the Y-axis represents epoch, and the Z-axis is the frequency. The first, middle and right columns present the results obtained by training the neural networks on balanced CIFAR-10, imbalanced CIFAR-10 without and with whitening, respectively. We can see that the main difference between the balanced and imbalanced tasks is that features learned on imbalanced dataset are more correlated than those on the balanced dataset, e.g., higher correlation coefficients and more singular values are nearly zero.

We give the results of ResNet-32 trained on balanced CIFAR-10, imbalanced CIFAR-10 with and without whitening operation in Fig 2. From the first row of sub-figure (b), we can see that the features learned on imbalanced dataset are more correlated than those on the balanced dataset. Previous works LeCun et al. (2002); Pezeshki et al. (2021); Zhu et al. (2023) have proved that: 1) The highly correlated features can produce gradient starvation and network degeneracies; 2) The high channel correlation can lead to feature redundancy. The singular value distributions in the second row (b) also demonstrate that high correlation reduces feature diversity, which brings more singular values closer to zero. Therefore, a technique designed to encourage the learning of a more diverse set of features by effectively decorrelating the learned representation is necessary.

We also provide more visualization results in the appendix, which includes visualizations on more datasets (CIFAR-10-LT, CIFAR-100-LT, iNaturalist-LT) based on different network architectures (ResNet-10, ResNet-32, ResNet-110, EfficientNet, DenseNet). All the provided results are consistent with the conclusion of the main paper, i.e., as the imbalance ratio increases, the features of the last hidden layer exhibit higher correlation.

## 3.2 Channel Whitening

**Channel Whitening vs. DBN Huang et al. (2018).** Previous work used whitening operation for general image classification to decorrelate features, which is called Decorrelated Batch Normalization (DBN) Huang et al. (2018). Our channel whitening method differs from it in the following two aspects. Firstly, we employ a selective application of Whitening. While DBN Huang et al. (2018) replaced all batch normalization layers in ResNet He et al. (2016) with whitening, our proposed method selectively applies whitening only in the last hidden layer. This selective approach effectively alleviates the degenerate solution and significantly reduces both training and inference time, as demonstrated in Table 5. Secondly, our approach focuses on Channel Whitening instead of Group Whitening. DBN addressed the computational complexity and non-convergence of whitening by dividing the channels into different groups. However, group whitening fails to adequately decorrelate each channel. Hence, our method, referred to as channel whitening, specifically aims at achieving channel decorrelation and overcoming the limitations associated with group whitening. Our results in Table 1

Table 1: The effectiveness of our channel whitening on imbalanced classification by comparing with the DBN Huang et al. (2018) method. The experiments are conducted on CIFAR-10-LT dataset with imbalance factor 200.

| Method | Accuracy |
|---|---|
| ERM | 66.4 |
| DBN Huang et al. (2018) | 67.1 |
| DBN Huang et al. (2018) *w/* Last Layer - Group Whitening | 66.6 |
| Ours *w/* Last Layer - Channel Whitening | 72.3 |
| Ours - WhiteningNet | 76.4 |

demonstrate that DBN and its group whitening are ineffective for imbalanced classification tasks. In contrast, our proposed method utilizes whitening selectively, specifically in the last hidden layer, which helps alleviate the degenerate solution while significantly reducing both training and inference time. Our approach achieves improved performance with an accuracy of 72.3%.

In the following, we present our proposed channel whitening technique to scatter the batch samples and decorrelate the feature representation to avoid network degeneration. The whitening transformation $\phi(\cdot)$ is defined as:

$$
\begin{aligned}
\phi(\mathbf{X}) =& \Sigma^{-\frac{1}{2}}(\mathbf{X} - \mathbf{u} \cdot \mathbf{1}^{\mathrm{T}}), \\
\mu_c =& \frac{1}{B} \sum_{i=1}^{B} \mathbf{X}_{ci}, \\
\mathbf{\Sigma} =& \frac{1}{C}(\mathbf{X} - \mathbf{u} \cdot \mathbf{1}^{\mathrm{T}})(\mathbf{X} - \mathbf{u} \cdot \mathbf{1}^{\mathrm{T}})^{\mathrm{T}} + \epsilon\mathbf{I},
\end{aligned}
\tag{2}
$$

where $\mathbf{u} = [\mu_1, \mu_2, .., \mu_C]^{\mathrm{T}} \in \mathbb{R}^C$ is a column vector with dimension $C$, $\mathbf{1}$ is a column vector with all entries being 1, $\mathbf{\Sigma}$ is the covariance matrix of zero-mean $\mathbf{X}$, and $\epsilon > 0$ is a small positive number for numerical stability (preventing a singular $\mathbf{\Sigma}$), $\mathbf{\Sigma}^{-\frac{1}{2}}$ is the inverse square root of the covariance matrix.

The ZCA whitening compute $\mathbf{\Sigma}^{-\frac{1}{2}}$ through eigen decomposition: $\Sigma^{-\frac{1}{2}} = \mathbf{V}\Lambda^{-\frac{1}{2}}\mathbf{V}^{\mathrm{T}}$, where $\Lambda = \mathrm{diag}(\lambda_1, \lambda_2, .., \lambda_C)$ and $\mathbf{V} = [v_1, v_2, ..., v_C]$ are the eigenvalues and eigenvectors of $\mathbf{\Sigma}$, $i.e. \mathbf{\Sigma} = \mathbf{V}\Lambda\mathbf{V}^{\mathrm{T}}$. The above process means that the centered $X$ is rotated by $\mathbf{V}^{\mathrm{T}}$, scaled by $\Lambda^{-\frac{1}{2}}$, and then rotated by $\mathbf{V}$ again. In the inference, we use the moving averaged of $\mathbf{u}$ and $\mathbf{\Sigma}^{-\frac{1}{2}}$ (Eq. 2) from training for channel whitening.

After whitening, the means of the feature representation $\phi(\mathbf{X})$ become zeros and its covariance matrix is the identity matrix, which implies that all the features are uncorrelated. As shown in top row of Figure 2 (c), the correlation coefficients among different channels are all zeros, which means the features are decorrelated, i.e., the linear dependencies have been removed. The bottom row results in Figure 2 (c) show that the whitened features in the last hidden layer have more large singular values to avoid feature concentration. Notably, our proposed channel whitening approach is only integrated before the linear classifier, which assure minimal computational overhead (Tabel 5).

## 3.3 Converence Analysis of ZCA Whitening

Our experiments demonstrate that ZCA whitening always fails to converge, especially under extremely imbalanced conditions. The investigation conducted by Huang et al. (2020a) concluded that applying whitening over batch data leads to significant instability during the model training. Furthermore, the study found that this instability often hinders convergence of whitening. Based on the above discussion in Section 3.2, although the group whitening proposed by DBN Huang et al. (2018) can reduce batch stochastic, it is ineffective in imbalanced classification. Therefore, below we observe the performance of covariance statistics on imbalanced data sets and propose new solutions.

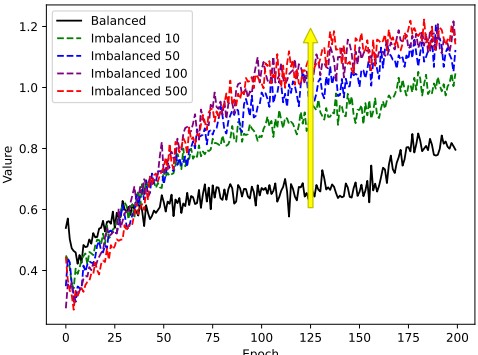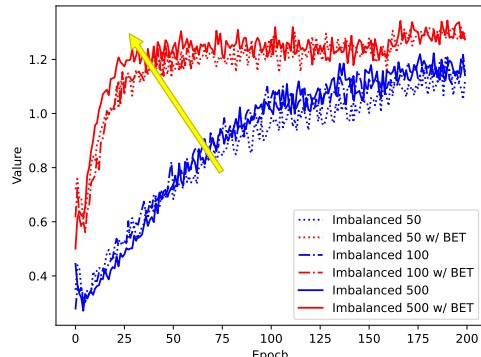

Figure 3: The visualization of batch covariance of last-layer hidden features before being fed into the classifier. The experiments are constructed on CIFAR-100-LT dataset using ResNet-32. We use "BET" to represent the proposed GRBS and BET approaches.

To analyze the stability of covariance matrix when the model is learned on imbalanced data, we define the following mini-batch covariance estimation metric:

$$E = \sum_{i=1}^{C} \mathbf{\Sigma}_{ii}, \tag{3}$$

where $\mathbf{\Sigma} \in \mathbb{R}^{C \times C}$ is the covariance matrix. $E$ represents the sum of the variances of all channels, i.e., a large value of $E$ means high stochasticity or instability in the covariance matrix.

As shown in Figure 3, we notice that on imbalanced classification, the mini-batch covariance statistics could be unstable, especially the imbalance ratio is large. This can affect the convergence and performance of whitening Huang et al. (2020a). Therefore, we would like to design a covariance-corrected module to obtain more stable and accurate batch statistics for avoiding non-convergence of channel whitening.

### 3.4 Covariance-corrected Modules

An obvious difference between neural networks trained on balanced and imbalanced data is the proportion of each class in the mini-batch samples. When the imbalance ratio is large, the widely used random sampler makes the tail classes difficult to participate in the mini-batch training. The inconsistency of sample categories in each batch causes the covariance statistics to always be unstable. Therefore, we would like to propose a new sampler and a novel training strategy to alleviate the above unstable problem, thus whitening can converge during training.

As shown in Figure 4, the proposed Group-based Relatively Balanced Sampler (GRBS) is divided into the following four steps:

- **[Group-based]** All $N$ categories are sorted according to their number of samples. The number of samples in each category is denoted to be $Q_i$, $i = 1, 2, ..., N$ ($\mathbf{Q} = [Q_1, Q_2, ..., Q_N]$). All sorted categories are equally divided into $G$ groups.

- **[Relatively Balanced]** In order to make the categories in each group relatively balanced, we select from $N$ sorted categories at equal intervals to form $G$ groups. To be precise, the $i$-th group is comprised from $\{i, i+G, i+2G, ..., i+(F-1)G\}$-th categories, $g = 1, 2, ..., G$. The number of samples of each category in group $i$ is denoted to be $\mathbf{Q_i} = [Q_i^1, Q_i^2, ..., Q_i^F]$, and we let their ratios be $\mathbf{R}_i = [R_i^1, R_i^2, ..., R_i^F]$. $F = \frac{N}{G}$ is number of category in each group.

- **[Sampling Probability]** We need a method that can automatically determine the sampling probability of each category in each group. For example, the sampling probabilities of $F$ categories in $i$-th group are $\mathbf{r_i} = [r_i^1, r_i^2, ..., r_i^F]$. Here, to get **relatively balanced** samples in each batch compared with random

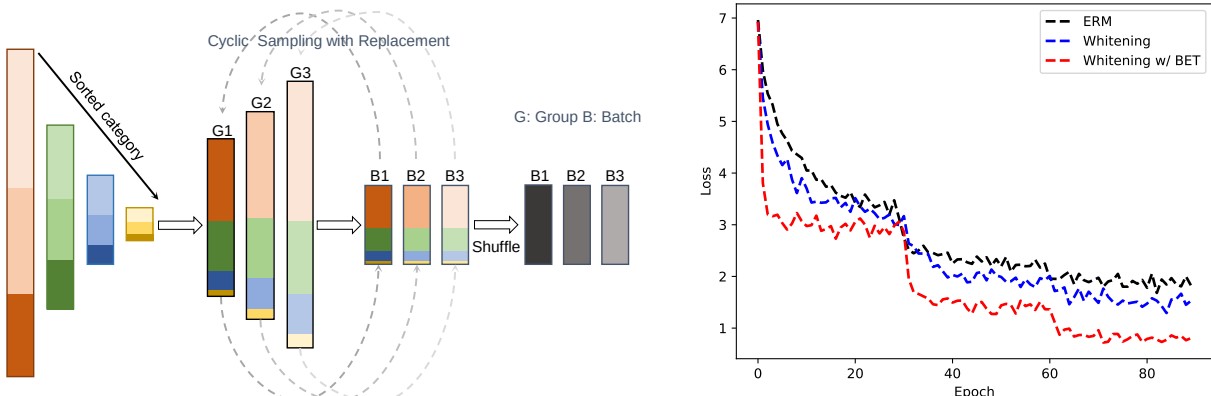

Figure 4: 1) The proposed group-based relatively balanced sampling method. The different colored rectangles represent different categories. 2) The visualization of training loss on iNaturalist-LT dataset. We use "BET" to represent the proposed GRBS and BET approaches.

sampling, we need to manually specify the sampling probabilities $\mathbf{r}_i^{F'}$ for the $F'$ tail classes in each group. Thus, the sampling probability $r_i^f$ of category $f$ in group $i$ can be calculated by the following equation:

$$r_i^f = \begin{cases} r_{min} & R_i^f \leq r_{min} \\ (1 - F'r_{min}) \times \hat{R}_i^f & R_i^f > r_{min} \end{cases} \tag{4}$$

where the new sampling probabilities $\hat{\mathbf{R}}_i \in \mathbb{R}^{F-F'}$ of the remaining $F - F'$ categories can be automatically calculated based on the ratio of their sample numbers. The setting of $r_{min}$ is used to increase the sampling probabilities of $F'$ tail categories. It is determined as follows:

$$r_{min} = \begin{cases} r_0 & S < 1 \\ S \times r_0 & 1 \leq S < S_0 \\ \frac{1}{F} & S >= S_0 \end{cases} \tag{5}$$

where $r_0$ is a minimum sampling probability, which is used to ensure that the smallest category of samples in the imbalanced dataset can be sampled. $S = \frac{Q_N/\alpha}{B/F}$ is a scale parameter, in which $Q_N$ is the number of samples in the smallest category of the entire imbalanced dataset, $\alpha$ is an adjustable parameter. $\frac{Q_N}{\alpha}$ is used to ensure the each sample of tail classes participate in training for every $\alpha$ epoch to prevent over-fitting. Because in the extremely imbalanced CIFAR-100-LT dataset, the smallest category may have only one sample. $\frac{1}{F}$ represents a class-balanced mini batch. $S_0$ is a scale threshold, which can divide the smallest category in the group into three categories. In our experiments, $S_0$ is fixed to 10.

- After the above steps, all categories are grouped and their sampling probabilities are recalculated. Since a larger sampling rate is specified for the tail class, the GRBS is a sampling process with replacement. Batch samples constructed using GRBS always come from categories in a certain group and are shuffled during use.

[**Batch Embedded Training**] Note that if we directly use the proposed GRBS instead of random sampler, the model could over-fit to the tail classes and it will also mistakenly weaken the representation learning of head classes. Therefore, we further propose a novel Batch Embedded Training (BET) strategy to eliminate these risks. To be precise, our strategy let the batches in the GRBS participate in the training intermittently (every $T$ iterations) in every epoch to promote the representation learning of the tail classes without sacrificing more learned knowledge on the head classes. We finally integrate this module together with GRBS into an end-to-end training scheme. The detailed steps of our proposed Whitening-Net in Algrithm 1. The ablation studies of hyper-parameters are provided in the appendix.

---

**Algorithm 1** Whitening-Net: An End-to-End Training Method for Imbalanced Classification

---

**Required Samplers:** Random Sampler with iterations $T_1$, GRBS Sampler with iterations $T_2$
**Required Models:** Initialized Backbone $f_{\theta 1}$ and Classifier $f_{\theta 2}$
**Required:** Inputs $\mathbf{X}$, Features $\mathbf{Z}$, Iteration threshold $T$, Channel Whitening $\phi$, Whitened features $\hat{\mathbf{Z}}$

1: **for** $t_1$=1 to $T_1$ **do**
2:     Extract features from random sampler $\mathbf{Z} = f_{\theta 1}(\mathbf{X})$
3:     Channel whitening $\hat{\mathbf{Z}} = \phi(\mathbf{Z})$
4:     Output logits $\hat{\mathbf{Y}} = f_{\theta 2}(\hat{\mathbf{Z}})$
5:     **if** $t_1/T = 0$ **then**
6:         **for** $t_2$=1 to $T_2$ **do**
7:             Extract features from GRBS sampler $\mathbf{Z} = f_{\theta 1}(\mathbf{X})$
8:             Channel whitening $\hat{\mathbf{Z}} = \phi(\mathbf{Z})$
9:             Output logits $\hat{\mathbf{Y}} = f_{\theta 2}(\hat{\mathbf{Z}})$
10:            Compute cross-entropy loss
11:            Update $f_{\theta 1}$ and $f_{\theta 2}$ by back-propagation
12:        **end for**
13:    **end if**
14:    Compute cross-entropy loss
15:    Update $f_{\theta 1}$ and $f_{\theta 2}$ by back-propagation
16: **end for**

---

Table 2: The details of imbalanced datasets.

| Dataset | # of classes | # of samples | Imbalance factor |
|---|---|---|---|
| CIFAT-10-LT | 10 | 50K | {10, 50, 100, 200} |
| CIFAT-100-LT | 100 | 50K | {10, 50, 100, 200} |
| ImageNet-LT | 1000 | 186K | 256 |
| iNaturalist-LT | 8142 | 437K | 500 |

## 4 Experiments

In this section, we firstly introduce the four imbalanced image classification datasets used for our experiments. Then we present some key implementation details of our experiments. After that, we present the comparison results with the state-of-the-art methods to show the superiority of our method. Finally, some ablation studies are given to highlight some important properties of our method.

### 4.1 Experimental Setup

**Datasets.** We perform experiments on three imbalanced datasets, including CIFAR-10-LT Krizhevsky et al. (2009), CIFAR-100-LT Krizhevsky et al. (2009), ImageNet-LT Deng et al. (2009) and iNaturalist-LT Van Horn et al. (2018). The details of datasets are presented in Table 2. Following prior work Cao et al. (2019), the long-tailed versions of CIFAR datasets are sampled from the balanced CIFAR by controlling the number of samples for each category. An imbalance factor $\gamma$ is used to present the ratio of training samples for the most frequent class and the least frequent class, $i.e., \gamma = \frac{N_{max}}{N_{min}}$. In our experiments, we set the imbalance factors as 10, 50, 100, 200 for CIFAR-10-LT and CIFAR-100-LT datasets. The large-scale ImageNet-LT consists of 115.8K training images from 1000 classes and the number of images per class is decreased from 1280 to 5. The iNaturalist-LT is a real-world, naturally long-tailed dataset, consisting of 437K training images from 8142 classes.

Table 3: Comparison with the state-of-the-art on CIFAR-10-LT and CIFAR-100-LT datasets. Best results of each column are marked in bold.

| Method | CIFAR-10 | | | | CIFAR-100 | | | |
|---|---|---|---|---|---|---|---|---|
| | 200 | 100 | 50 | 10 | 200 | 100 | 50 | 10 |
| *End-to-end training* | | | | | | | | |
| ERM | 66.4 | 71.2 | 77.4 | 86.8 | 34.4 | 38.6 | 43.8 | 56.7 |
| CB-CE | 68.8 | 72.7 | 78.2 | 86.9 | 35.6 | 38.8 | 44.8 | 57.6 |
| Focal Lin et al. (2017) | 65.3 | 70.4 | 76.8 | 86.7 | 35.6 | 38.4 | 44.3 | 55.8 |
| MW-Net Shu et al. (2019) | 67.2 | 73.6 | 79.1 | 87.5 | 36.6 | 41.6 | 45.7 | 58.9 |
| LDAM-DRW Cao et al. (2019) | 73.0 | 77.2 | 81.6 | 87.6 | 38.8 | 42.8 | 47.3 | 57.5 |
| Casual Tang et al. (2020) | - | 80.6 | 83.6 | 88.5 | - | 44.1 | 50.3 | 59.6 |
| LADE Hong et al. (2021) | | | - | | | 45.4 | 50.5 | 61.7 |
| MFW Ye et al. (2021) | 75.0 | 79.8 | - | 89.7 | 41.1 | 46.0 | - | 59.1 |
| Hybrid-PSC Wang et al. (2021) | - | 78.8 | 83.9 | **90.1** | - | 45.0 | 48.9 | 62.4 |
| CMO Park et al. (2022) | - | - | - | - | - | 46.6 | 51.4 | 62.3 |
| *Decoupled training* | | | | | | | | |
| BBN Zhou et al. (2020) | - | 79.8 | 82.2 | 88.3 | - | 42.6 | 47.0 | 59.1 |
| Ours | **76.4** | **80.6** | **84.0** | 89.6 | **43.0** | **47.2** | **52.2** | **62.9** |

Table 4: Comparison with the state-of-the-art on ImageNet-LT and iNaturalist datasets. Best results of each column are marked in bold.

| Method | ImageNet-LT | | | | iNaturalist-LT | | | |
|---|---|---|---|---|---|---|---|---|
| | Many | Medium | Few | All | Many | Medium | Few | All |
| *End-to-end training* | | | | | | | | |
| ERM | 55.1 | 22.4 | 2.2 | 32.3 | 55.7 | 45.5 | 40.6 | 44.6 |
| CB-CE | 56.8 | 25.7 | 3.2 | 34.6 | 46.2 | 49.8 | 47.2 | 47.5 |
| LDAM Cao et al. (2019) | 51.0 | 25.2 | 4.9 | 32.4 | 45.7 | 49.3 | 50.9 | 49.6 |
| CMO Park et al. (2022) | 50.2 | 33.5 | 21.2 | 38.3 | 43.6 | 51.7 | 54.7 | 52.3 |
| *Decoupled training* | | | | | | | | |
| NCM Kang et al. (2020) | 42.6 | 33.0 | 20.1 | 35.0 | 30.2 | 38.1 | 41.6 | 38.6 |
| cRT Kang et al. (2020) | 51.4 | 38.4 | 22.5 | 41.0 | 49.6 | 51.5 | 50.4 | 50.9 |
| LWS Kang et al. (2020) | 49.3 | 39.0 | 23.9 | 40.7 | 44.3 | 51.0 | 52.9 | 51.1 |
| Ours | 53.6 | 38.7 | 21.2 | **41.5** | 49.3 | 53.4 | 53.8 | **53.2** |

### 4.2 Implementation Details

All the experiments are implemented by Pytorch 1.7.0 on a virtual workstation with 11G memory Nvidia GeForce RTX 2080Ti GPUs. All the experiments are reproduced based on the released codes.

**Long-tailed CIFAR.** For both long-tailed CIFAR-LT-10 and CIFAR-LT-100 datasets, following most of the existing work, we use ResNet-32 He et al. (2016) as backbone to extract image representation. SGD optimizer is adopted to optimize model with momentum of 0.9, weight decay of 0.0002. The initial learning rate is set to 0.1 and is decreased to 1/10 of its previous value on the 160-th and 180-th epoch of the total 200 epochs. The batch size is set to 128.

**ImageNet-LT and iNaturalist-LT.** We use ResNet-10 He et al. (2016) as backbone model. SGD optimizer with momentum of 0.9, weight decay of 0.0005. The initial learning rate is set to 0.2 and is decreased to 1/10 of its previous value for every 30 epochs in the total 90 epochs. The batch size is set to 512. We adhere to the approach outlined in Liu et al. (2019) for reporting accuracy across three class splits:: Many-shot (more than 100 images), Medium-shot (20-100 images) and Few-shot (less than 20 images).

Table 5: Model efficiency (s/per epoch) on imbalanced datasets based on ResNet-32, ResNet-10. The time is the average of all the training epochs. The model is trained with 200 epochs on CIFAR-10 datset, 90 epochs on ImageNet-LT and iNaturalist-LT datasets.

| Method | CIFAR-10 | | iNaturalist-LT | | ImageNet-LT | |
|---|---|---|---|---|---|---|
| | Training | Inference | Training | Inference | Training | Inference |
| ERM | 3.42 | 1.22 | 814.2 | 50.3 | 113.3 | 18.2 |
| ERM w/ Whitening | 3.98 | 1.48 | 817.8 | 54.2 | 116.2 | 22.0 |

Table 6: The results are used to verify the effectiveness of our proposed GRBS and BET. The testing accuracy is obtained from CIFAR-100-LT dataset. "CB" represents class-balanced sampling.

| Method | Imbalance factor | | | |
|---|---|---|---|---|
| | 200 | 100 | 50 | 10 |
| ERM | 34.4 | 38.6 | 43.8 | 56.7 |
| w / CB | 28.2 | 31.2 | 38.5 | 53.4 |
| w / GRBS | 32.1 | 33.2 | 41.0 | 54.6 |
| w / GRBS & BET | 36.5 | 39.7 | 45.0 | 57.8 |
| w / Whitening | 41.2 | 43.5 | 47.8 | 59.6 |
| w / Whitening & CB | 35.3 | 38.1 | 43.6 | 55.8 |
| w / Whitening & GRBS | 39.3 | 40.7 | 46.1 | 58.6 |
| w / Whitening & GRBS & BET | **43.0** | **47.2** | **52.2** | **62.9** |

## 4.3 Main Results

In this section, we present results to demonstrate the effectiveness of our proposed Whitening-Net method by comparing with the state-of-the-art baselines, including the end-to-end and decoupled training methods. The experimental results on hyperparameters are in the appendix.

### 4.3.1 Results on CIFAR-LT-10/100

As shown in Table 3, our proposed method Whitening-Net achieves better performance than the decoupled training method BBN Zhou et al. (2020) over different imbalanced factors by a large margin, especially in CIFAR-100-LT. Compared with the state-of-the-art methods, Hybrid-PSC Wang et al. (2021) obtains 90.1% on CIFAR-10-LT with imbalanced factor 10, but it performs worse on CIFAR-100-LT. Therefore, these results verify that with our Whitening-Net framework, end-to-end training can achieve better performance and whitening can be used to escape from the degenerated solutions.

### 4.3.2 Results on ImageNet-LT and iNatunalist-LT

The results illustrated in Table 4 show that our method can achieve 53.2% of the overall performance on iNaturalist-LT, which is better the second best results 51.1% achieved by decoupled method LWS Kang et al. (2020). Compared with the state-of-the-art methods, CMO Park et al. (2022) performs worse than our Whitening-Net although CMO is trained with 400 epochs and AutoAugmentation.

### 4.3.3 Computational Cost of Whitening

We analyze the computational cost of whitening on different datasets and network architectures. As shown in Table 5, the training time increases by about 4 seconds per epoch on iNaturalist-LT dataset, and the inference time on CIFAR-10 dataset is increases by only 0.26 second. The results show that computational cost added by whitening approach is very small.

Table 7: The hyper-parameters of GRBS and BET on different datasets. $G$ is the group number, $r_0$ is the basic sampling probability, $T$ is the iteration interval for BET.

| Hyper-parameter | $G$ | $r_0$ | $\alpha$ | $T$ |
|---|---|---|---|---|
| CIFAT-10-LT | 1 | 0.05 | 2 | 60 |
| CIFAT-100-LT | 10 | 0.01 | 2 | 30 |
| ImageNet-LT | 100 | 0.01 | 2 | 60 |
| iNaturalist-LT | 815 | 0.01 | 2 | 200 |

Table 8: Top 1 accuracy by varying group number $G$ on iNaturalist-LT dataset.

| Group number $\#G$ | Many | Medium | Few | All |
|---|---|---|---|---|
| 200 | 47.2 | 53.3 | 55.8 | 53.0 |
| 400 | 48.4 | 53.1 | 55.0 | 53.1 |
| 800 | 49.3 | 53.4 | 53.8 | **53.2** |
| 1000 | 48.6 | 52.8 | 54.2 | 53.0 |

### 4.3.4 Effectiveness of GRBS & BET

As illustrated in Figure 4, our proposed whitening approach integrated with BET make the training loss smaller, which means the model can jump out of the degenerate solution. The visualizations on Figure 3 show that our proposed covariance-corrected modules make the covariance structure more stable during training, thus avoiding non-convergence. The testing accuracy verified in Table 6 also demonstrate the effectiveness of our proposed GRBS and BET, i.e., it can reinforce the capability of channel whitening. In addition, their combination performs better than the class-balanced sampling, because the GRBS let the tail classes participate in more iterations without affecting the representation learning of head classes, and the BET training strategy can make the GRBS avoid over-fitting to tail classes.

## 5 Ablation Studies on Hyper-parameters

In this section, we provide some ablation studies on the hyper-parameters of GRBS and BET. All the experiments are constructed based on the proposed Whitening-Net on the large scaled iNaturalist-LT dataset.

Table 7 presents all the hyper-parameters of GRBS and BET. In our experiments, $S_0$ and $\alpha$ are fixed across all the imbalanced datasets. We make batches sampled from GRBS always have a fixed number of categories, which makes $G$ easy to compute.

**Hyper-parameters of GRBS.** The hyperpameters of GRBS include group number $G$, minimum sampling ratio $r_0$. As shown in the Table 8, the performances under different hyper-parameter $G$ of GRBS are similar, i.e., the proposed Whitening-Net is not sensitive to the choice of hyper-parameters. At the same time, we can control the classification accuracy of different shots by selecting different group number $G$, e.g., small group number $G$ means that the class imbalance in each group is more serious, and more tail classes will be sampled to alleviate the imbalance in each group, because we fix the minimum sampling ratio $r_0$ of the tail classes. The experimental results in Table 9 also demonstrate that the minimum sampling ratio $r_0$ can control the classification accuracy of the samples in each shot, and a larger $r_0$ will enhance the model's ability to recognize tail classes.

**Hyper-parameters of BET.** The parameter of iteration interval $T$ denotes that in each epoch, the samples in the proposed GRBS participate in training after $T$ iterations of random sampler, i.e., samll $T$ means that the samples in GRBS participate in more training to augment the representation learning of tail class. The experimental results illustrated in Table 10 demonstrate that the proposed BET training strategy is robust to hyper-parameter $T$.

Table 9: Top 1 accuracy by varying minimum sampling ratio $r_0$ on iNaturalist-LT dataset.

| Sampling ratio #$r_0$ | Many | Medium | Few | All |
|---|---|---|---|---|
| 0.002 | 48.8 | 53.3 | 53.4 | 52.8 |
| 0.01 | 49.3 | 53.4 | 53.8 | **53.2** |
| 0.02 | 48.9 | 53.3 | 54.9 | 53.1 |
| 0.03 | 48.3 | 53.4 | 55.6 | 53.1 |
| 0.04 | 46.9 | 53.2 | 56.7 | 53.0 |

Table 10: Top 1 accuracy by varying iteration interval $T$ ($G = 800, r_0 = 0.01$) on iNaturalist-LT dataset.

| Iteration interval #$T$ | Many | Medium | Few | All |
|---|---|---|---|---|
| 100 | 49.1 | 53.4 | 53.9 | 53.0 |
| 200 | 49.3 | 53.4 | 53.8 | **53.2** |
| 300 | 49.4 | 53.4 | 53.6 | 53.1 |

## 6 Conclusion

In this paper, we first identify that the highly correlated feature representations fed into the classifier is the key factor causing the failure of end-to-end training scheme on imbalanced classification. Then, we propose a simple yet effective framework Whitening-Net, which integrates channel whitening into the end-to-end training to scatter the features and thus avoid representation collapse. Another contribution of this paper is we propose two covariance-corrected modules to get more accurate and stable batch statistics to avoid non-convergence and reinforce the capability of whitening. Our results demonstrate that with our whitening technique, end-to-end training scheme can avoid model degeneration. Although our proposed Whitening-Net has shown considerable improvements on the benchmarks of imbalanced learning, we hope to explore novel approach to replace the SVD computation of whitening transformation.

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

In this appendix, we provide more visualization results in Section A to prove that the model does have network degeneration phenomenon on imbalanced data, where the experiments come from different datasets (CIFAR-10-LT, CIFAR-100-LT and iNaturalist-LT), different backbones (ResNet-10, ResNet-32, ResNet-110, EfficientNet-B0 and DenseNet121).

## A Visualization for Network Degeneration

### A.1 Visualization on All Hidden Layers

In Figure 5, we visualize the channel-wised singular value distributions on different layers of ResNet-32. From Figure 5 (a) and (b), we observe that the main difference between the features learned in the imbalanced and balanced datasets is that the last intermediate hidden layer, i.e., the features fed into the classifier, learned on the imbalanced dataset have a significantly larger amount of nearly zero-valued singular values, which implies that these features are highly correlated. This feature representation collapse would make the training intractable and finally leads to degenerated solutions.

### A.2 Visualization on CIFAR-100-LT

As shown in Figure 6, we provide a visualization results on CIFAR-100-LT with imbalanced factor 200, in which we can draw the same conclusion as in the paper, i.e. the features trained on an imbalanced dataset have larger correlation coefficients, and the proposed ZCA whitening approach can alleviate this problem.

### A.3 Visualization on More Network Architectures

We also construct experiments on CIFAR-10-LT dataset using more different backbones, e.g., ResNet-110, EfficientNet-B0 and DenseNet121, to prove the conclusion in the main paper. As shown in Figure 7, 8 and 9, the correlation coefficient values of the last layer features increases with the imbalance ratio of the dataset.

### A.4 Visualization on iNaturalist-LT

We also present visualization results on large scaled iNaturalist-LT dataset to show the effectiveness of our proposed ZCA whitening. As shown in Figure 10, the result in top figure shows that when trained with ERM model, more than 95% of the singular values are smaller than 10. In contrast, we can see that when trained with Weighting-Net, the learned features have more large-valued singular values, implying that the features are effectively decorrelated.

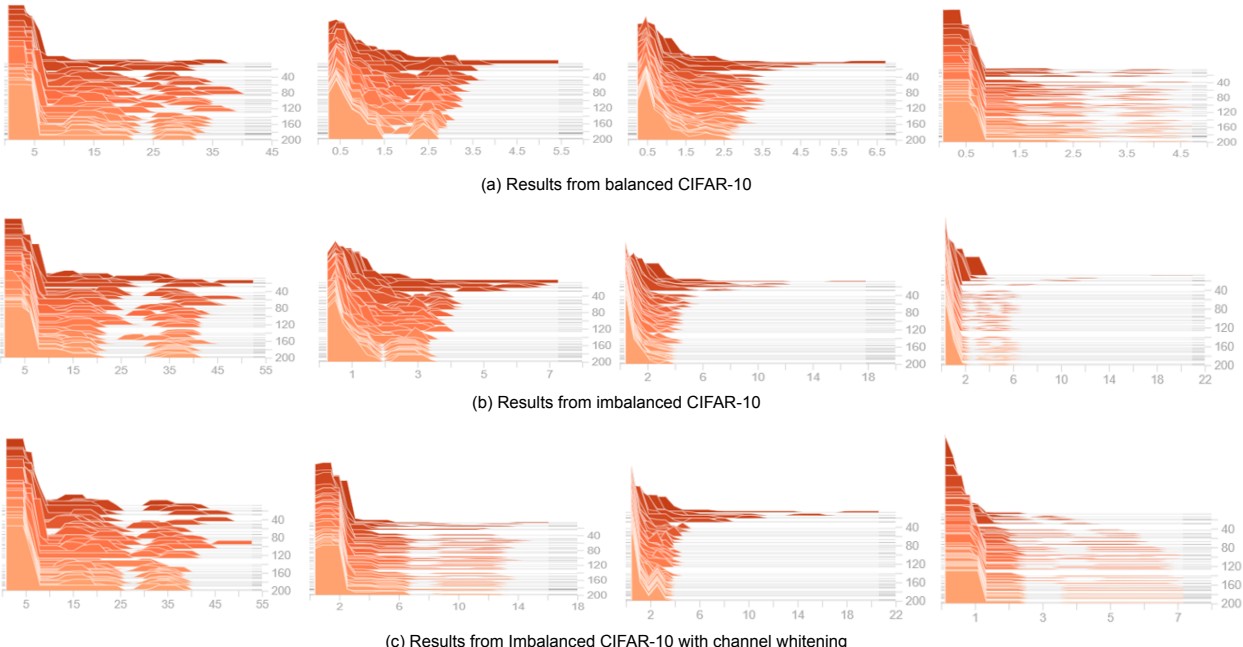

Figure 5: Singular value histograms of features on different layers (The sub-figures of (a), (b) from left to right are: Layer_1, Layer_2, Layer_3 and Layer_p, where "p" denotes pooling. The last sub-figure on (c) is Layer_p after whitening transformation.) of ResNet-32 using end-to-end training. The first, middle and bottom rows present the results on balanced CIFAR-10, imbalanced CIFAR-10 and imbalanced CIFAR-10 with whitening, respectively. The vertical axis in each figure stands for the training epoch. We can see that the main difference between the balanced and imbalanced tasks is that a large amount of the singular values of the features fed into classifier (i.e., the last column) in the imbalanced task are nearly zero, which implies that these features are highly correlated. The bottom row demonstrates that our whitening can effectively decorrelate these features since the features have more large singular values.

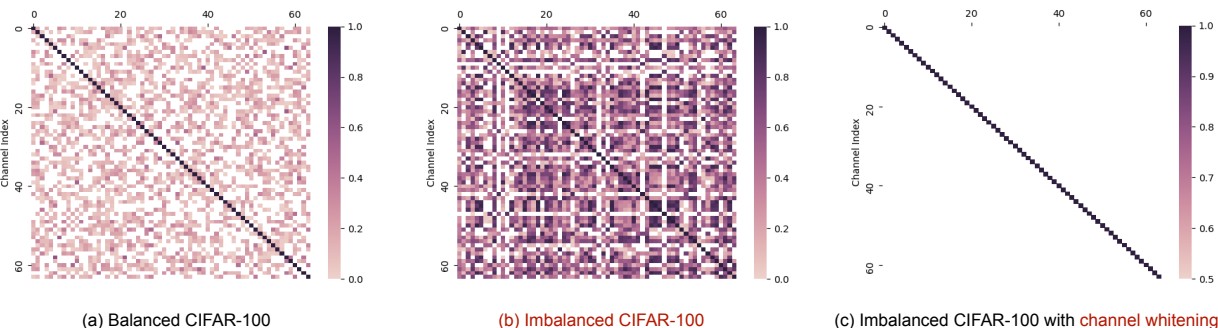

Figure 6: The correlation coefficients between channel-wised features fed into the classifier at the last epochs. The experiments are constructed on CIFAR-100-LT datasets using ResNet-32.

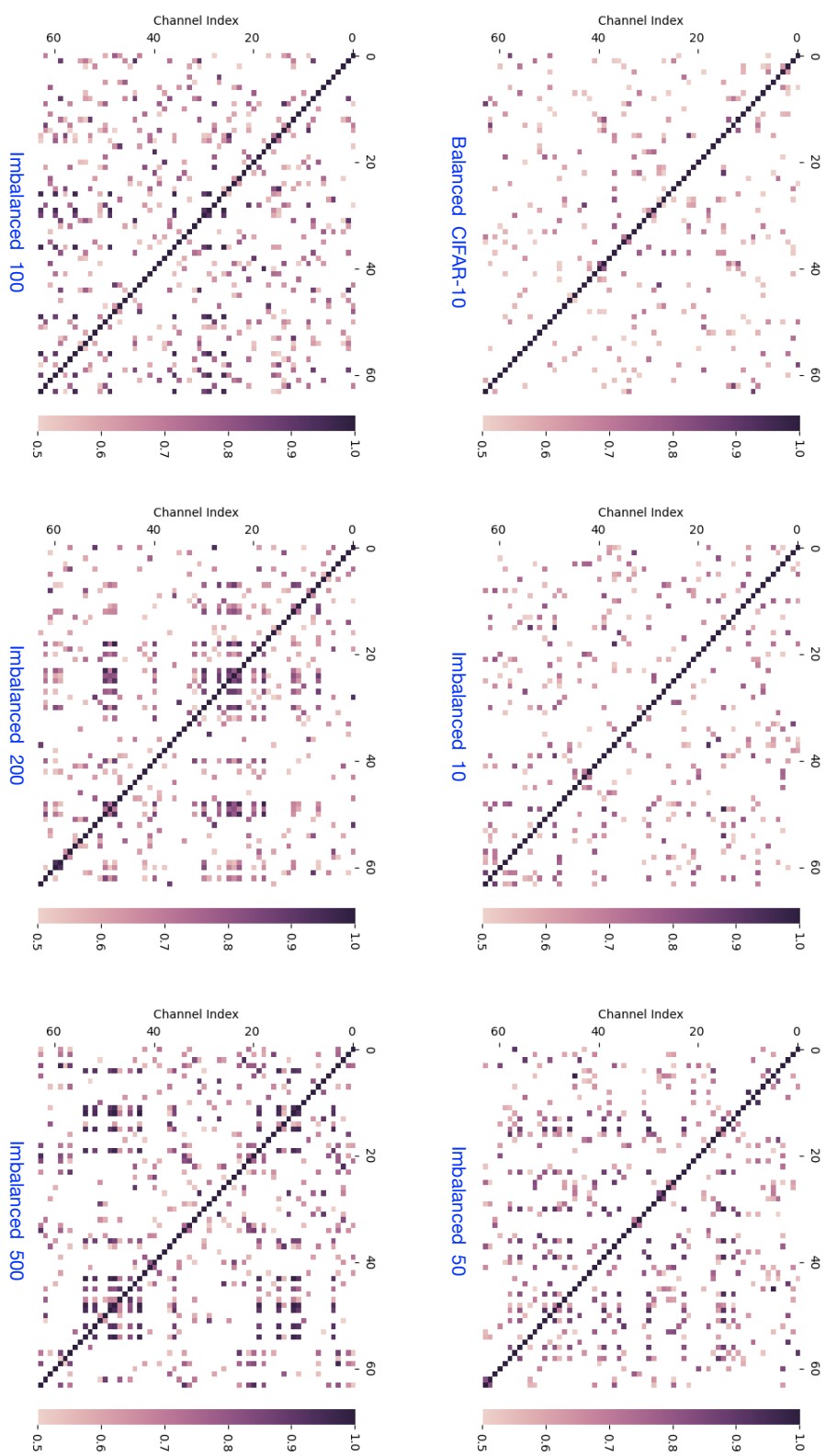

Figure 7: The correlation coefficients between channel-wised features fed into the classifier at the last epochs. The results are obtained on CIFAR-10-LT dataset using ResNet-110.

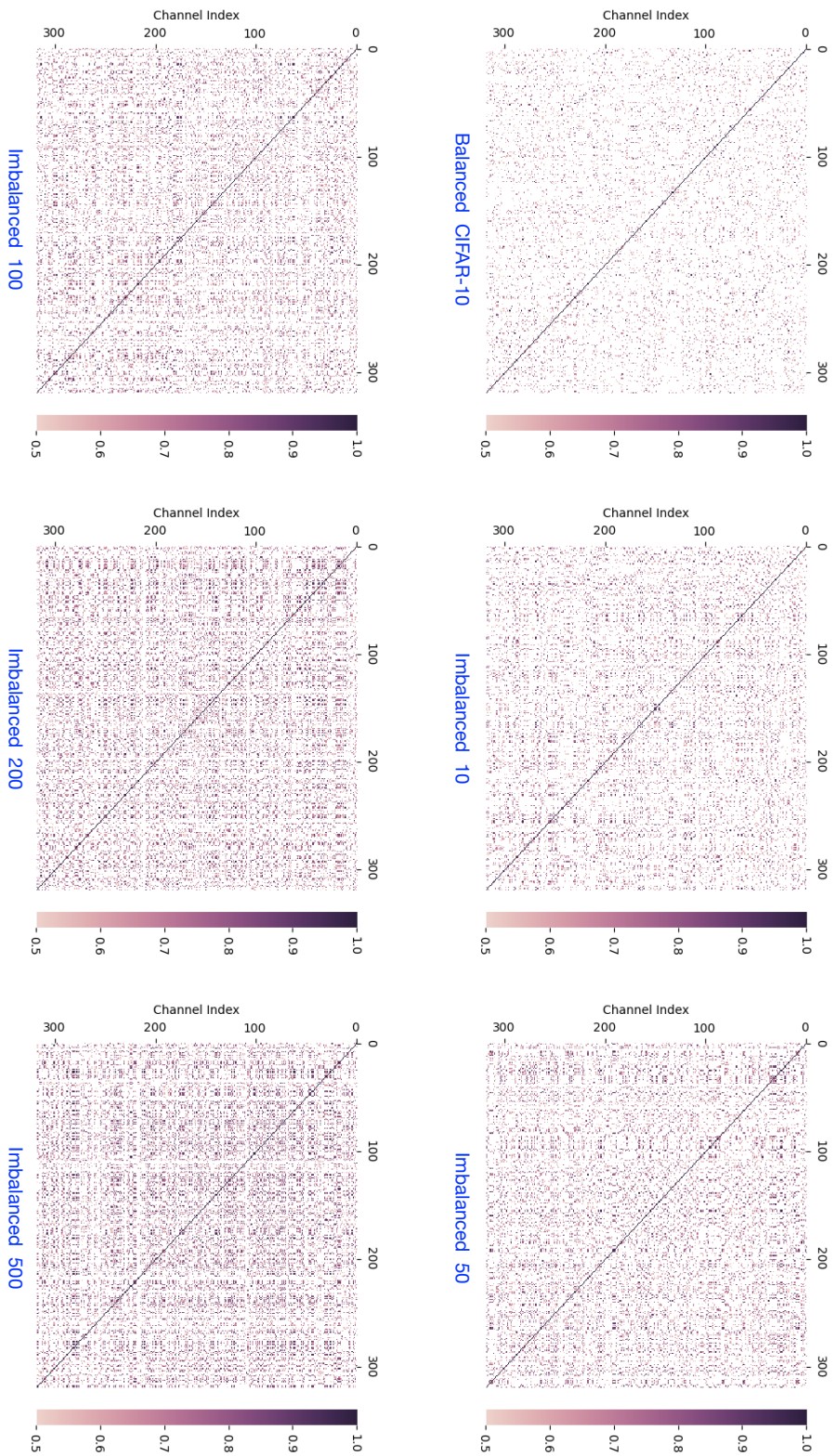

Figure 8: The correlation coefficients between channel-wised features fed into the classifier at the last epochs. The results are obtained on CIFAR-10-LT dataset using EfficientNet-B0.

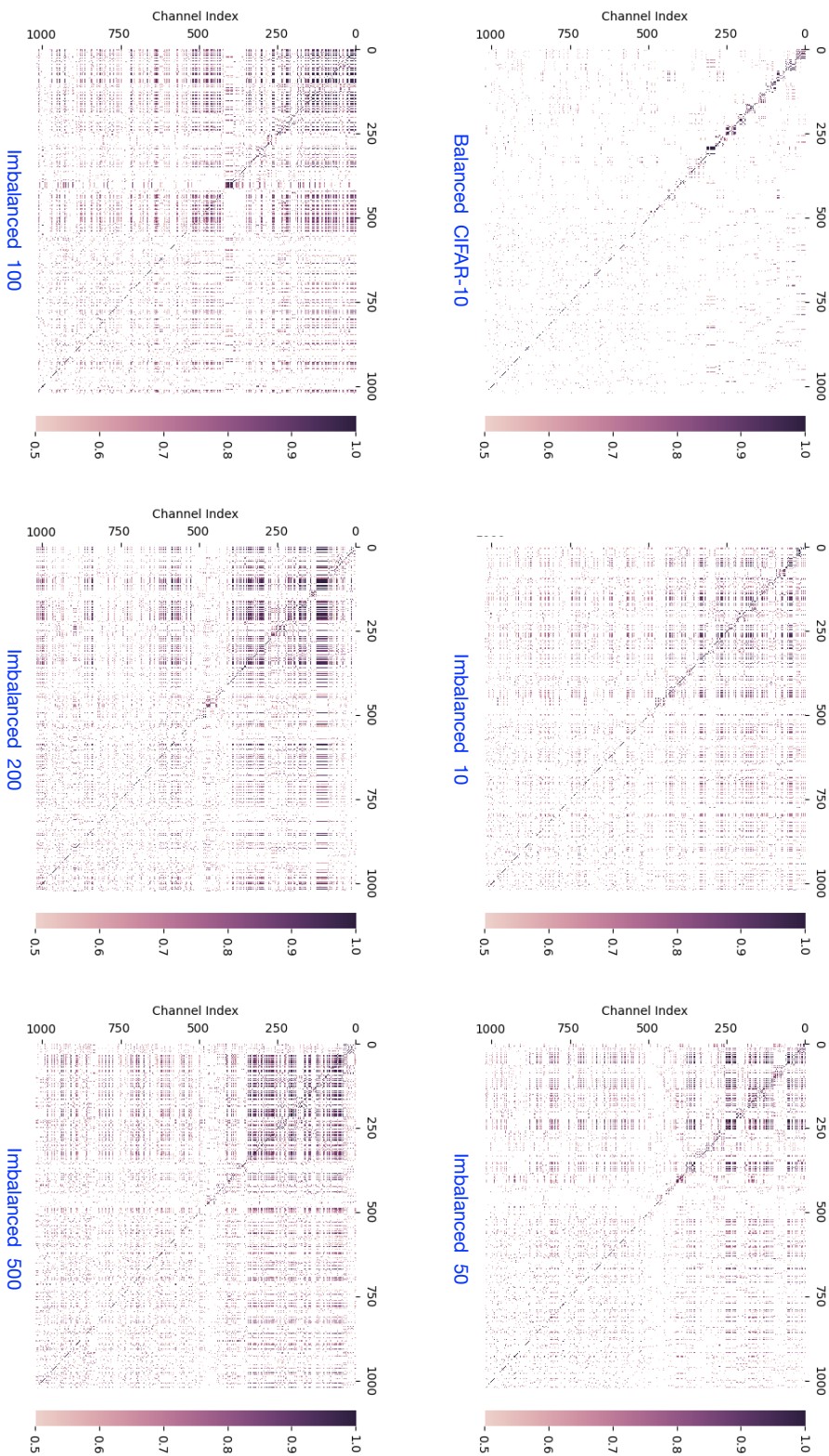

Figure 9: The correlation coefficients between channel-wised features fed into the classifier at the last epochs. The results are obtained on CIFAR-10-LT dataset using DenseNet-121.

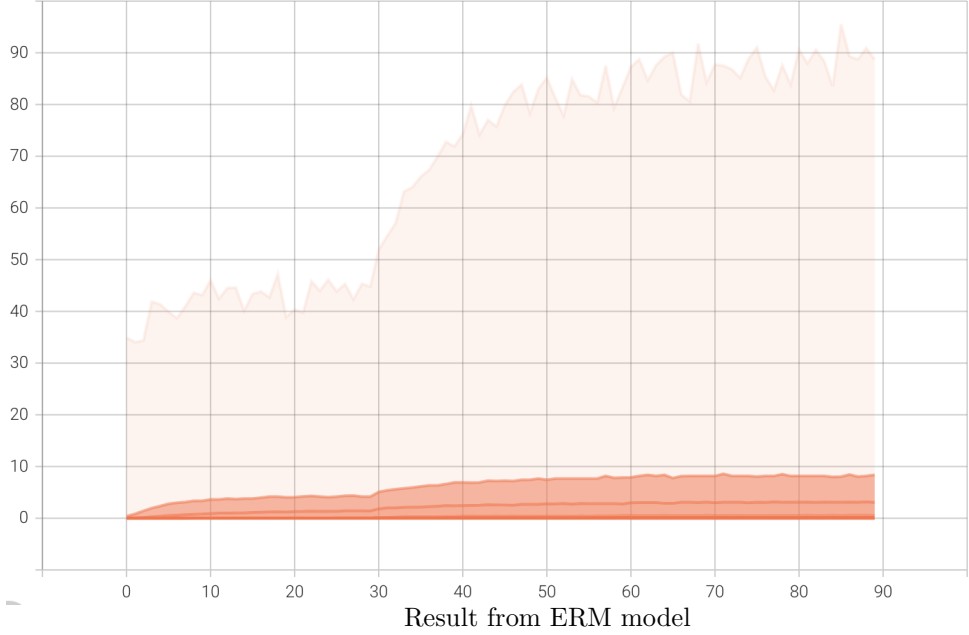

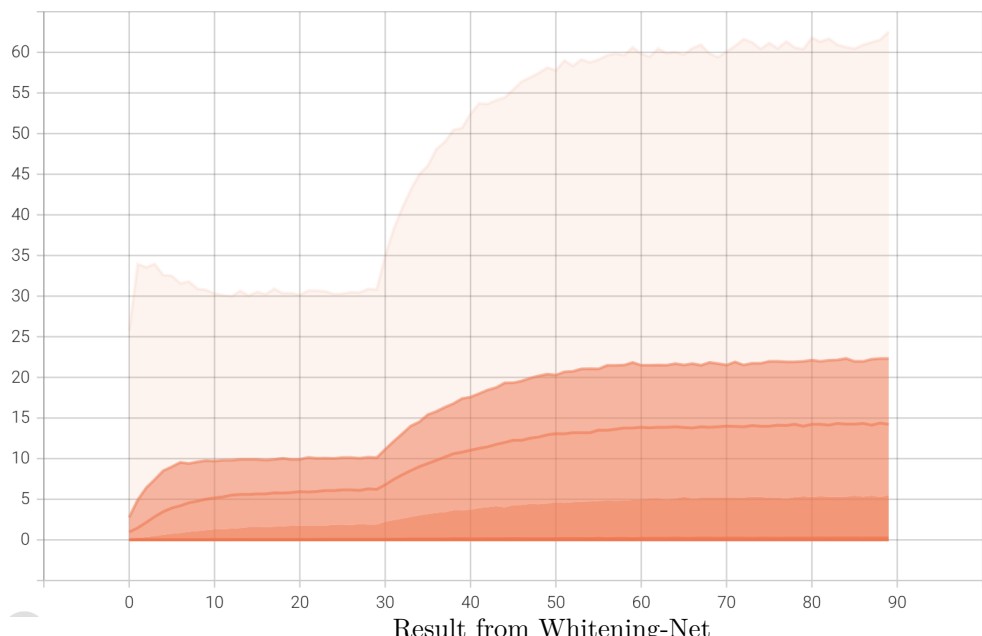

Figure 10: Singular value distributions of features fed into the classifier. The experiments are constructed on iNaturalist-LT dataset using ResNet-10. $x$-axis stands for number of epoch, $y$-axis is the singular value. The curves from the top to the bottom are maximum-value, 99.7% quantile, 95% quantile, 68% quantile and the minimum-value, respectively. The result in top figure shows that when trained with ERM, more than 95% of the singular values are smaller than 10. **In contrast, we can see that when trained with Weighting-Net, the learned features have more large-valued singular values, implying that the features are effectively decorrelated.**

