# OpenReview forum: "Covariance-corrected Whitening Alleviates Network Degeneration on Imbalanced Classification"
_TMLR — Withdrawn by Authors_

### Review · Reviewer_2Noj · 2024-09-30

**Summary Of Contributions:**

This paper propose a neural network (NN) training model to tackle class-imbalance problem in image classification task.

With the observation that the NN classifiers trained with standard ERM tends to output features with highly correlation between channels, a novel NN training model is proposed, which consists of:
1. A novel GRBS sampler that sample images in a more class-balanced way
2. A novel "Whitening" operation on last layer feature to de-correlate the channel-wise features

The author(s) then report experiment results on CIFAR10/100, ImageNet-LT and iNaturalist datasets to demonstrates the effectiveness of the model.

**Audience:**

Yes

**Claims And Evidence:**

No

**Requested Changes:**

I think computer vision journal/conference is a better place for this paper to go. For other comments, pleas see strengths and weakness.

**Strengths And Weaknesses:**

Pros:
1. Most part of the paper are well written
2. Visualizations looks interesting.

Cons:
1. Novelty: the GRBS sampler is essentially a re-sample technique and the Whitening operation is essentially a normalization technique. The combination of the two approach does not looks novel to me.
2. May not be the right venue: most of this work focus on empirical analysis. Almost no theoretical analysis are provided. Therefore I think this work may not be suitable for the machine learning venue.
3. Experiment results are mainly compared with 2020~ works, what about more recent works? Why are they not compared?
4. Clear typo in section title: "3.3 Converence Analysis of ZCA Whitening", what is "Converence "?

---

> ### Comment · Reviewer_2Noj · 2024-10-01
> **Some clarification**
>
> Thanks for the reply, let me clarify the points:
>
> Clarification:
>
> 1. The contribution of "the observation of the correlated channels in feature space for network trained with class-imbalanced data" is strong if the paper could at least provide some theoretical discussion of :
>      1) why class-imbalance problem cause such phenomenon;
>      2) how this phenomenon influence the performance.
>
> Illustration of such phenomenon looks good but not a strong evidence of novelty.
>
> 2. The contribution of the proposed model is strong if it could
>      1) compare with other methods and shows superior performance;
>      2) provides reasons that why some other methods are not compared.
>
> Superior performance over some of the existing method looks good but not a strong evidence of superior performance.
>
> Moreover, the idea of grouping classes in a balanced way has appeared in SuperDisco [CVPR 2023], which learns a group of class-balanced "super-class".
>
> This paper is mostly empirical, when less theoretical discussions are provided, I would expect more evidence in terms of superior performance to justifies it's contribution, comparing with recent papers like "Curvature-Balanced Feature Manifold Learning for Long-Tailed Classification"[CVPR 2023], SuperDisco [CVPR 2023], "Balanced Product of Calibrated Experts for Long-Tailed Recognition"[CVPR 2023]) would be a good choice.
>
> When both theoretical evidence and empirical justification are not strong enough, I cannot give a recommendation of acceptance.

---

### Review · Reviewer_T8HC · 2024-10-29

**Summary Of Contributions:**

This paper studied the problem of long-tailed image classification, where the underlying labeling distributions may be unbalanced across different classes. The paper starts by asking the question of what causes the poor performance of ERM training for imbalanced classification. The identified issue is that the highly correlated features fed into the classifier makes the failure of end-to-end ERM training.

In response to this issue, the paper proposes a Whitening-Net integrating the whitening transformation into the model to decorrelate the features before being fed into the classifier.

There are two covariance correction modules, Group-based Relatively Balanced Batch Sampler, and Batch Embedded Training, within the proposed method.
- The first involves four steps: Grouping, balancing, biased sampling, and recalculating sampling probabilities.
- The second involves "let the batches in the GRBS participate in the training intermittently (every T iterations) in every epoch to promote the representation learning of the tail classes without sacrificing more learned knowledge on the head classes."

To validate the approach, experimental results are reported on CIFAR-10-LT, CIFAR-100-LT, ImageNet-LT, and iNaturalist-LT. The results justify the utility of the approach, in terms of empirical improvements over the baselines.

**Audience:**

Yes

**Broader Impact Concerns:**

I have no concerns about this aspect.

**Claims And Evidence:**

Yes

**Requested Changes:**

- To ensure that the proposed algorithm is not just ad-hoc, and can generalize to different types of problems. I would like to see if the authors could add some theoretical justification to their algorithm. One reference from this literature is:

Kaidi Cao, Colin Wei, Adrien Gaidon, N. Aréchiga, and Tengyu Ma. Learning imbalanced datasets with label- distribution-aware margin loss. In NeurIPS 2019

- To further validate that the proposed algorithm is indeed advancing the state-of-the-art, I would like to know what the results would be when applied to Waterbirds, CeleA, and CivilComments-WILDS. For references about these datasets, see below:

Zhang, M., Sohoni, N.S., Zhang, H.R., Finn, C. and Re, C., 2022, June. Correct-N-Contrast: a Contrastive Approach for Improving Robustness to Spurious Correlations. In International Conference on Machine Learning (pp. 26484-26516). PMLR.

Kirichenko, P., Izmailov, P. and Wilson, A.G., Last Layer Re-Training is Sufficient for Robustness to Spurious Correlations. In The Eleventh International Conference on Learning Representations. 2023


Based on how well the authors respond to these two requested changes, I will re-evaluate this work when making the final recommendation.

**Strengths And Weaknesses:**

S1) The problem of long-tailed classification is an important one in practice.

The proposed approach is relatively simple, and would be easy implement (if the code is open sourced).

S2) The paper is fairly easy to follow, although there are multiple typos, and the citation format is messed up (many are missing the brackets).

S3) The experiments are solid.

W1) There is no theoretical/rigorous justification for the algorithm. This makes the empirical results relatively mysterious and difficult to interpret.

W2) The datasets tested are a bit restricted. There are many datasets nowadays featuring correlated features. To ensure the generalizability of the findings, the experiments need to be conducted on more recent datasets with imbalanced labels.

---

### Review · Reviewer_DwVH · 2024-11-17

**Summary Of Contributions:**

This paper develops tools for whitening data in class imbalance scenarios. It proposes two methods for whitening data in extremely unbalanced settings: the Group-based Relatively Balanced Batch Sampler (GRBS) and Batch Embedded Training (BET). To address instability in naive whitening approaches, the authors combine resampling and whitening to build more stable covariance estimates. The methods are evaluated on CIFAR-LT-10/100, ImageNet-LT, and iNaturalist-LT, showing improvements over other whitening and resampling approaches.

**Audience:**

Yes

**Broader Impact Concerns:**

None noted

**Claims And Evidence:**

Yes

**Requested Changes:**

1. **Additional Comparisons:**
   - Add comparisons with **DBN**, other whitening methods, and spectral approaches tailored for imbalanced datasets.
   - Provide a detailed analysis of how and when the different components of your method contribute to performance improvements in various settings.

2. **Stronger Justification:**
   - Provide clearer theoretical or empirical justification for combining CB, GRBS, and BET. Including synthetic datasets or controlled experiments would help demonstrate the utility of these combinations.

3. **Inclusion of Recent Work:**
   - Discuss and compare your methods with recent spectral methods (e.g., Ma, 2022; Kaushik, 2024).
   - Highlight any unique contributions relative to feature decorrelation techniques in SSL.

4. **Presentation Improvements:**
   - Correct typographical errors (e.g., "CIFAT" in Table 2, "iNatunalist"). See above.

---

**Strengths And Weaknesses:**

### Strengths
- **Relevance:** The paper addresses an important issue of class imbalance, which is a major challenge in machine learning.
- **Performance Gains:** The proposed methods demonstrate improvements over naive whitening approaches on multiple datasets, providing empirical support for their utility.

---

### Weaknesses
1. **Lack of Clear Justification:**
   - The paper combines multiple strategies (e.g., CB, GRBS, BET) to achieve good performance, but the rationale for why these specific components are necessary for whitening is unclear.
   - For example, in **Table 6**, adding CB or GRBS individually reduces performance compared to whitening alone. Improvements are observed only when CB, GRBS, and BET are combined. This raises questions about the necessity of each component. Rigorous testing and clearer explanations of these interactions are needed.

2. **Missing Related Work:**
   - The paper overlooks recent work leveraging the spectrum of latent features to address class imbalance (e.g., Ma, 2022; Kaushik, 2024). Comparisons with these methods would provide important context and highlight the novelty of the proposed approach.

3. **Method Design:**
   - The method appears somewhat ad hoc, stitching together several small techniques without a clear justification. Additional evaluations across diverse settings and datasets are needed to demonstrate how these components interact and their combined efficacy.

4. **Insufficient Baseline Comparisons:**
   - The comparisons in **Table 1** focus on resampling and whitening combinations but lack evaluations against other resampling-based approaches or spectral methods. Including these would strengthen the paper’s claims. Similarly, in the LT evaluations, we are missing context about how whitening approaches or DBN would perform.
- How do your resampling methods (BET, GRBS) perform when combined with DBN (whitening across layers, and/or group level)?

5. **Clarity and Presentation:**
   - The use of "channels" versus "features" in the data model is inconsistent and unclear.
   - Several typos reduce readability, e.g., **Table 2** ("CIFAT"), **Section 4.3.1** ("iNatunalist").
   - The motivation describes DBN, but later comparisons are made with BBN. Including DBN in the results would provide consistency and clarity.
 - The term "experiential risk minimization" is used without definition and is not standard. This needs to be clarified for readers.
- Can you clarify the use of "channels" versus "features" in your data model? The terminology seems inconsistent.

6. **Relation to SSL Methods:** How do your methods relate to feature decorrelation approaches like VICReg in self-supervised learning? This connection would be interesting to explore.

---

### References
- **Ma, 2022:** Ma, Yanbiao, et al. *Delving into semantic scale imbalance*. arXiv preprint arXiv:2212.14613, 2022.
- **Kaushik, 2024:** Kaushik, Chiraag, et al. *Balanced Data, Imbalanced Spectra: Unveiling Class Disparities with Spectral Imbalance*. arXiv preprint arXiv:2402.11742, 2024.

---

### Review · Reviewer_55Lu · 2024-11-17

**Summary Of Contributions:**

The paper addresses the issue of class imbalance in image classification, which significantly affects the performance of deep recognition models. The authors identify a network degeneration dilemma caused by high linear dependence among the features inputted into the classifier. To tackle this, they propose an approach based on ZCA whitening called Whitening-Net, which integrates ZCA whitening before the linear classifier to normalize and decorrelate batch samples. However, extreme class imbalance causes fluctuations in batch covariance statistics, impeding the whitening operation. To address this, the authors propose two covariance-corrected modules: Group-based Relatively Balanced Batch Sampler (GRBS) and Batch Embedded Training (BET). These modules help achieve more accurate and stable batch covariance, reinforcing the whitening capability. Empirical evaluations on benchmark datasets like CIFAR-LT-10/100, ImageNet-LT, and iNaturalist-LT validate the effectiveness of their approaches.

**Audience:**

Yes

**Claims And Evidence:**

No

**Requested Changes:**

* Quotations in brackets
* Define "Experiential Risk Minimization" in the introduction
* Add "?" at the end of *what causes the poor performance of end-to-end Experiential Risk Minimization (ERM) model training for the imbalanced classification*
* *Prior works LeCun et al. (2012) demonstrated that the good features should be decorrelated and have same covariances to avoid producing degeneracies. To this end, we propose ... * : 1) avoiding linear correlation (with whitening net) does not guarantee that features are correlated 2) only bivariate correlation are considered (with whitening net) 3) same variance rather than covariance.
* Reference figure 1 in the body of the paper
* Add some recent work to related works
* *Re-sampling. *the oversampling methods augment the tail classes by duplicating samples and they could result in over-fitting*. This is true only for simple oversampling (by bootstrap resampling), but synthetic data generation methods, as SMOTE, are precisely the answer to this shortcoming.
* I suggest renaming *3. Method* with the algorithm's name
* Add appendix reference and shortcut (use "ref{}")
* *While DBN Huang et al. (2018) replaced all batch normalization layers in ResNet He et al. (2016) with whitening,* why *He et al. (2016)* here ?
* The *3. method* section incorporates results from the experiments section; it could be clearer to have a theoretical section and a different numerical illustration section.
* *Converence Analysis of ZCA Whitening*: *Convergence* and it looks more like a behavioral analysis than a convergence analysis.
* Figure 3 : change y-axis label with "batch covariance"
* Using subgraphics and subcaptions would make the paper more clearer
* *the mini-batch covariance statistics could be unstable* covariance for imbalanced scenarios is higher but does not seem more unstable (volatile)
* 3.4: in my opinion, this is the most important part of the paper in terms of contributions.
   * All sorted categories are equally divided into G groups.* no need to mention it here, as it is specified below
   * *Relatively Balanced* : $F$ is used before its definition
   * *we select from N sorted categories*: we *divide* no ?
    * *the i-th group* add $G_i$
    * $g$ is not used
    * define ratios $R_i$
    * It would be useful to add a numerical illustration (with the scheme): what are the values for N, F, G, etc. in Figure 4 ? [I'd like to point out that this part isn't clear to me yet, a lot of notations]
* *4. Experiments*
    * Recall the metric studied in the results: accuracy?
    * It would be interesting to use the balanced accuracy as a performance metric (or other metrics specifically designed for imbalanced classification)
    * Table 3: recall that 200, 100, 50, 10 are imbalanced factors (tables and results must be understandable with their legends - without looking for explanations in the text)
* *samll T means that the samples in GRBS* : small

**Strengths And Weaknesses:**

Strengths:
* Performance analysis of the ZCA whitening method in unbalanced classification and justification with feature correlation.
* Full experimentation with 4 real data sets, several competitors, different imbalance factors, three class separations, ablation studies, computational analysis, hyperparameter sensitivity analysis
* Overall clarity of the paper

Weaknesses:
* using whitening in imbalanced learning does not seem to be a new concept (e.g. 3 recent papers [1], [2], [3] and [8] + [9] which seems introduce a similar method + [10] ?)
* Related works are not recent: less than 2021 (cf papers [4], [5], [6], [7] below)
* With all due respect to the proposal, the application of the channel whitening (in the last hidden layer and instead of channels group) is not a methodological novelty, just a difference in the way whitening is applied.
* Methodological justifications are based entirely on empirical and numerical considerations. No explanation is given. For example: *This selective approach effectively alleviates the degenerate solution and significantly reduces both training and inference time, as demonstrated in Table 5.* Could you justify the term *effectively alleviates the degenerate solution* ? or in conclusion *we first identify that the highly correlated feature representations fed into the classifier is the key factor causing the failure of end-to-end training scheme on imbalanced classification*: it would be really beneficial to explain this phenomenon.
* 3.4 : The proposal includes improving whitening by balancing batches. What is the difference between the proposed method (GRBS) and a weighting of observations for batch selection or a weighting of observations in the loss function ?


[1] Wang, Z. and Wang, H. (2024). Variational imbalanced regression: Fair uncertainty quantification via
probabilistic smoothing. Advances in Neural Information Processing Systems, 36.
[2] Yang, Y., Zha, K., Chen, Y., Wang, H., and Katabi, D. (2021a). Delving into deep imbalanced regression.
In International Conference on Machine Learning, pages 11842–11851. PMLR.
[3] Ding, Y., Jia, M., Zhuang, J., and Ding, P. (2022). Deep imbalanced regression using cost-sensitive
learning and deep feature transfer for bearing remaining useful life estimation. Applied Soft Computing,
127:109271.
[4] Zhang, Y., Kang, B., Hooi, B., Yan, S., and Feng, J. (2023c). Deep long-tailed learning: A survey. IEEE
Transactions on Pattern Analysis and Machine Intelligence.
[5] Moukpe, J. (2023). Survey on Imbalanced Data, Representation Learning and SEP Forecasting. arXiv preprint arXiv:2310.07598.
[6] Chen, W., Yang, K., Yu, Z., Shi, Y., & Chen, C. L. (2024). A survey on imbalanced learning: latest research, applications and future directions. Artificial Intelligence Review, 57(6), 1-51.
[7] Yokoi, S., Bao, H., Kurita, H., & Shimodaira, H. (2024). Zipfian Whitening. arXiv preprint arXiv:2411.00680.
[8] Sharma, S., Bellinger, C., Krawczyk, B., Zaiane, O., & Japkowicz, N. (2018, November). Synthetic oversampling with the majority class: A new perspective on handling extreme imbalance. In 2018 IEEE international conference on data mining (ICDM) (pp. 447-456). IEEE.
[9] Ermolov, A., Siarohin, A., Sangineto, E., & Sebe, N. (2021, July). Whitening for self-supervised representation learning. In International conference on machine learning (pp. 3015-3024). PMLR.
[10] Li, J., Wang, Y., Zi, Y., & Zhang, Z. (2021). Whitening-Net: A generalized network to diagnose the faults among different machines and conditions. IEEE transactions on neural networks and learning systems, 33(10), 5845-5858.

---

### Note · Authors · 2024-12-16

**Comment:**

I thank all the reviewers for their comments. For a variety of reasons, I have chosen to withdraw this paper.

**Withdrawal Confirmation:**

I have read and agree with the venue's withdrawal policy on behalf of myself and my co-authors.